# A Cross-Talk about Radioresistance in Lung Cancer—How to Improve Radiosensitivity According to Chinese Medicine and Medicaments That Commonly Occur in Pharmacies

**DOI:** 10.3390/ijms241311206

**Published:** 2023-07-07

**Authors:** Paulina Nowak, Iwona Bil-Lula, Mariola Śliwińska-Mossoń

**Affiliations:** 1Scientific Club of Specialized Biological Analyzes, Faculty of Pharmacy, Wroclaw Medical University, Borowska 211A, 50-556 Wroclaw, Poland; 2Department of Medical Laboratory Diagnostics, Division of Clinical Chemistry and Laboratory Hematology, Faculty of Pharmacy, Wroclaw Medical University, Borowska 211A, 50-556 Wroclaw, Poland; iwona.bil-lula@umw.edu.pl

**Keywords:** radiotherapy, lung cancer, traditional Chinese medicine

## Abstract

Lung cancer is one of the most common cancers in the population and is characterized by non-specific symptoms that delay the diagnosis and reduce the effectiveness of oncological treatment. Due to the difficult placement of the tumor, one of the main methods of lung cancer treatment is radiotherapy, which damages the DNA of cancer cells, inducing their apoptosis. However, resistance to ionizing radiation may develop during radiotherapy cycles, leading to an increase in the number of DNA points of control that protect cells from apoptosis. Cancer stem cells are essential for radioresistance, and due to their ability to undergo epithelial–mesenchymal transition, they modify the phenotype, bypassing the genotoxic effect of radiotherapy. It is therefore necessary to search for new methods that could improve the cytotoxic effect of cells through new mechanisms of action. Chinese medicine, with several thousand years of tradition, offers a wide range of possibilities in the search for compounds that could be used in conventional medicine. This review introduces the potential candidates that may present a radiosensitizing effect on lung cancer cells, breaking their radioresistance. Additionally, it includes candidates taken from conventional medicine—drugs commonly available in pharmacies, which may also be significant candidates.

## 1. Introduction

Lung cancer leads the statistics in terms of morbidity and mortality among all malignant tumors. Taking into account the usually late diagnosis of the disease, the incidence rate decreases dramatically. Radiotherapy is one of the basic methods for lung cancer treatment in monotherapy by reducing the tumor mass. It is also used in combination with anticancer drugs. The development of radioresistance during therapy is a serious problem that scientists and doctors are constantly trying to prevent. Therefore, compounds that could increase the pro-apoptotic and genotoxic effectiveness of radiotherapy are needed. Generally, potential candidates can be selected from medicinal raw materials that have a long history of use in traditional medicine in different parts of the world. They are used as treatments for various diseases or alongside well-known conventional drugs in off-label use. Nowadays, one of the topical themes is traditional Chinese medicine, which has been used in Asia for almost 2000 years. The aim of this review is to introduce the methods that enhance the effectiveness of radiation therapy. For better clarity in this review, radiosensitizers were divided into two sections. One section describes radiosensitizers with a natural origin used in traditional Chinese medicine, and the second section describes well-known drugs used in conventional medicine.

## 2. Lung Cancer: Epidemiology, Morphology and Symptoms

Lung cancer remains one of the most aggressive cancers in the world. In 2022, the Cancer Center Statistics, a part of the American Cancer Society, documented 23,6740 new cases of lung cancer morbidity in the US alone [1]. The last report, called GLOBOCAN, from 2022 estimated 19.3 million new cancer cases, of which 2,206,771 were lung cancer cases. Thus, lung cancer was the second most common cause of cancer (11.4% of total cases) after female breast cancer (11.7% of total cases) in 2020. Therefore, lung cancer is a serious epidemiological problem that has not yet been fully resolved. Moreover, lung cancer was the most mortal cancer (1,796,144 cases) [2]. Considering the histological conditions as well as therapeutic possibilities and treatment prognosis, lung cancer can be divided into two groups: small-cell carcinoma (SCLC), which represents 13% of total cases, and non-small-cell carcinoma (NSCLS), which includes 83% of total cases [3]. At an early stage of progression, lung cancer is asymptomatic; therefore, most cases are diagnosed at an advanced stage, which reduces the effectiveness of treatment. The main symptoms are not specific, such as hoarseness, cough (often with blood), and low-grade fever. Almost 60% of new cases are diagnosed during the metastatic phase in other tissues, while 22% of cases are diagnosed with metastasis only to the lymph nodes. The efficiency of the treatment is correlated with the stage at diagnosis. The five-year survival in the distant metastatic phase is estimated at 5.2%, in the regional stage at 30.8%, and in the localized stage at 57.4% [4].

### Molecular Classification of Lung Cancers

The extremely heterogeneous nature of lung cancer has become more apparent over the past decade. In general, advanced lung cancer is an aggressive malignancy with a poor prognosis. However, the discovery of multiple molecular mechanisms underlying the development, progression, and prognosis of lung cancer has opened up new opportunities for targeted therapy and improved outcomes.

Most lung cancers are non-small-cell lung cancers (NSCLCs), with the most common morphological subtypes being adenocarcinoma (AC), squamous cell carcinoma (SCC), and large cell neuroendocrine carcinoma (LCNEC) [5]. This purely morphological classification has been challenged in recent decades, as ACs can be further subdivided into distinct molecular subtypes. These molecular subtypes are defined by a single change in the oncogenic factor, including gene mutations, rearrangements, and amplifications [5].

There is considerable variability in the incidence of oncogenic driver changes, which occur mainly in AC, are very rare in SCC, and are generally more common in Asian populations [6,7]. Molecular subtyping has become very important as genotype-based therapy (“targeted therapy”) is now the standard of care for a significant subset of patients with advanced and metastatic NSCLC (stage IV), which has led to improved outcomes.

The discovery of activating epidermal growth factor receptor (EGFR) mutations as the first targeted and predictive oncogenic driver change in lung cancer has profoundly changed the diagnosis and therapeutic landscape of lung AC [8]. The treatment of EGFR-mutated NSCLC with EGFR tyrosine kinase inhibitors (TKIs) is a model of biomarker-based therapy and highlights the importance of establishing a predictive molecular biomarker for personalized therapy in the treatment of solid tumors [5].

With significant progress in the treatment options for patients with advanced, inoperable NSCLC, prognostic biomarkers, including morphological subtype, PD-L1 status, and the results of targeted oncogenic changes of drivers, are a prerequisite for making decisions about the best possible therapy, as they are directly related to the choice of a specific treatment. While the current guidelines focus on testing genes such as EGFR, ALK, ROS1, and BRAF for approved drugs, they recognize the rapidly changing treatment landscape and suggest performing broader molecular profiling, including genes such as MET, RET, NTRK, KRAS, and HER2 [9]. The Federal Drug Administration (FDA) recently approved drugs targeting NSCLC with MET exon 14 skipping changes and RET and NTRK 1/2/rearrangements, respectively [5].

Genome sequencing of a large sample of NSCLC could facilitate the identification of new molecular targets and the development of new therapies to improve the outcomes of lung cancer patients.

In the May 2016 issue of Nature Genetics, Campbell et al. reported the results of exome sequence analysis and copy number profiles of 1144 lung cancers, including 660 AC and 484 SCC [6]. They identified 38 significantly mutated genes in lung AC and 20 significantly mutated genes in lung SCC. Fourteen genes were found to be mutated in different types of cancer but not identified in a separate analysis. With this large dataset, low-frequency changes in critical oncogenic pathways such as RTK/RAS/RAF pathway genes (SOS1 and RASA1) and Rho kinase signaling genes (VAV1 and ARHGAP35) were identified. Only six genes, such as TP53, RB1, ARID1A, CDKN2A, PIK3CA, and NF1, were significantly mutated in both tumor types, suggesting that somatic lesions may have different oncogenic potential in different cellular contexts. In addition, new significantly mutated genes have been reported, including PPP3CA, DOT1L, and FTSJD1 in lung AC; RASA1 in SCC lung; and KLF5, EP300, and CREBBP in both tumor types. The new amplification peaks include MIR21 in lung ADC, MIR205 in lung SCC, and MAPK1 in both [5].

These findings contribute to the characterization of a highly heterogeneous neoplastic process in two major subtypes of lung cancer that reflect individual genetic predisposition, different progenitor cells, and stochastic acquisition of mutational events that are later selected as oncogenic factors.

Small-cell lung cancer (SCLC) is a malignant neoplasm with an unusual proliferative and invasive capacity. Due to the lack of effective therapies, it has a very poor clinical prognosis. In recent years, studies on cells, animal models, and tumor samples have enabled the identification of SCLC molecular subtypes, the discovery of unique biological and clinical features, and the proposition of potential specific therapeutic targets for the various subtypes. Advances in gene sequencing technologies and the reduction of sequencing costs have led to the increased importance and use of individualized sequencing to identify mutation profiles in lung cancer patients. This facilitates individualized and precise therapy [10]. In addition, the development of molecularly targeted drugs for common mutations in lung cancer is advancing rapidly. Patients with mutations in the EGFR, ALK, RAS-MAPK, RET, MET, and BRAF genes are subjected to targeted therapy, which results in significant improvement in patients’ survival [11,12,13]. The molecular mechanisms of mutational evolution in SCLC are complex. Therefore, molecular subtypes cannot be identified without comprehensive sequencing data. The rapid development of gene sequencing technology and individualized, precise therapy has led to an increase in the use of next-generation sequencing (NGS) in SCLC [14]. Another study suggested that Notch may be a tumor suppressor in SCLC, and the Notch family was affected by genomic changes in 1/4 of SCLC cases [15]. Secondary mutations in oncogenes driven by oncogenic TP53 have been found to be mainly responsible [13,14,15,16]. TP53 mutations were present in approximately 90% of SCLC samples; most of them were missense mutations (63% of TP53 mutations) that affected DNA binding functions. The Rb1 gene was completely deleted or mutated in 64% of SCLC patients [15]. Two tumor suppressors, p53 and Rb, were found to be simultaneously inactivated in the vast majority of SCLC samples. The MYC gene family includes MYC, MYCL, and MYCN. MYC is able to activate the Notch pathway to dedifferentiate tumor cells, thereby promoting the progression of SCLC.

Some studies have indicated that SCLC arises from monoclonal proliferation driven by mutations in neuroendocrine stem cells [17]. Based on the expression patterns of neuroendocrine markers such as neural cell adhesion molecule 1 (NCAM1/CD56), chromogranin A (CHGA), synaptophysin (SYP), and gastrin-releasing peptide (GRP) in tumor cells, SCLC can be categorized into high neuroendocrine subtypes (NE) and low/no neuroendocrine (low NE/non-NE) [18,19].

Based on neuroendocrine subtypes, researchers realized that there are large differences in molecular mutations and expression patterns in different SCLC samples. Based on analytical results from human primary SCLC tissues, patient-derived xenografts, mouse genetic models (GEMMs) and SCLC cell lines, Rudin et al. [19] summarized previous classification methods and identified differences in the expression of the main transcription factors: ASCL1, NEUROD1, YAP1 and POU2F3, SCLC into four molecular subtypes as follows: SCLC-A, SCLC-N, SCLC-Y and SCLC-P [18,19]. Three of them have ASCL1, NEUROD1 and POU2F3 gene changes, respectively, and the fourth subtype lacks ASCL1, NeuroD1 and POU2F3 mutations [19].

The latest version of the WHO pathology guidelines did not introduce the mutation-based molecular subtypes of NSCLC and SCLC [19]. Although mutations can only be detected by sequencing, abnormal biological behaviors of tumor cells caused by mutations can provide clues through histopathological changes. Therefore, a histopathological combination of tumors in conjunction with molecular mutations is expected to be used in clinical practice. Molecular subtyping based on gene mutations is complementary to conventional histopathology.

## 3. All about Radiation Therapy and Radioresistance

Radiotherapy is the first-line treatment for hard-to-reach cancers, for instance, oral cancer, lung cancer, and head and neck cancers. However, radiation therapy can also be used before surgery to reduce tumor mass, either as a complementary method with cytostatic agents or after surgery.

Ionizing radiation provided directly to the tumor mass initiates DNA damage, finally inducing cancer cell death due to apoptosis, autophagy, necroptosis, ferroptosis, and senescence [20]. DNA damage is induced by two different mechanisms: directly by inducing single and double-strand breaks or indirectly by free radical formation [21]. During a few weeks of lung cancer radiotherapy, ionizing radiation of 1.8–2 Gy per day at cycle for five days was used, followed by two days off. However, increasing radioresistance constitutes a huge challenge for oncologists.

Cancer stem cells (CSC) are considered to be one of the main reasons for the development of radioresistance in tumor masses. Similarly, to normal stem cells, CSC possess unique properties such as self-renewal, dedifferentiation, proliferation, and multipotency, but they have lost control of these processes. After irradiation, CSC presented increased DNA damage checkpoints, stimulation of DNA repair mechanisms, and increased free radical elimination. Furthermore, radiation can activate pathways associated with the EMT (epithelial–mesenchymal transition) process such as Wnt/β-catenin, Notch, and Hedgehog, leading to enhanced survival of cancer cells [22]. The tumor mass microenvironment (CSC niche) plays a crucial role in the maintenance of CSC plasticity. It has been confirmed that radiation of the TME (tumor microenvironment) using CAFs (cancer-associated fibroblasts) reorganizes the niche in severe actions, for instance, cytokine secretion, recruitment of immune cells, and up-regulation of maintenance stemness such as Nanog factors [23].

Apart from the CSC concept, gene mutations are also involved in the development of acquired radioresistance. Mutations in various oncogenes such as EGFR, K-RAS, and tumor suppressors such as TP-53 have been observed in NSCLS. The EGFR and EGFR-ligands are integral factors inducing cell proliferation and activation of pro-survival signaling pathways such as PI3K/AKT, MAPK, activators of the transcription signaling cascade, and several transducers. Blockade of EGFR pathways constitutes a potential target for radiosensitizing therapy [24]. Another well-known mutation is the K-RAS gene mutation, which participates in carcinogenesis in almost 30% of all cases. Therefore, pro-cancer mechanisms of action are still unknown, but they are probably associated with cancer stem cells [25].

Recent publications suggest that long non-coding RNA may also induce radioresistance through tumor microenvironment remodeling by changing miRNAs, STAT, Wnt, EZH2, PTEN, and PI3K/Akt signaling pathways [26,27,28].

Radiosensitizers are chemical compounds characterized by a high affinity to electrons. They possess large redox potential and contribute to the replacement of oxygen in hypoxic cells. They are nitroheteroaromatic compounds containing the nitro group -NO_2_, so-called nitroimidazoles: metronidazole (flagyl), misonidazole (Ro-07-0582), pimonidazole (Ro 3-8799), ethanidazole (SR 2508) [29]. Other well-known radiosensitizers are taxans: docetaxel and paclitaxel, tested in pre-clinical and clinical trials [30].

## 4. What If Traditional Chinese Medicine Is an Answer to Radiosenstization?

Most studies on herbs in traditional medicine that may act as radiosensitizers present curcumin and berberine. Curcumin is a polyphenol isolated from Curcuma longa and is mainly used in digestive diseases [31,32]. Additionally, curcumin acts as a protective agent in various diseases of the cardiac, neurological, and immune systems [33]. Berberine is a kind of alkaloid isolated from *Phellodendron chinense* Schneid that was used in traditional medicine for the treatment of infectious diseases. On the other hand, searching for new radiosensitizers is not limited to traditional medicine. In practice, more and more drugs have been tested off-label in different conditions. The properties of individual compounds are summarized in Table 1.

### 4.1. Oridonin

One of the newest, potentially radiosensitizing candidates is oridonin, a diterpenoid mainly isolated from *Rabdosia rubescens* (*Isodon rubescens*) and other plants from the Isodon group. *Rabdosia rubescens* has been used in traditional Chinese medicine. The dry aerial part of these plants, called *Donglingcao* in Chinese, was described in the Chinese Pharmacopeia as an agent with anti-inflammatory and detoxifying properties that relieves pain. Traditionally, it has been used in acute and chronic pharyngitis, bronchitis, tonsillitis, and insect bites [67]. Many studies indicate it has a wider range of therapeutic effects. Oridonin demonstrates antibacterial, antioxidant, and immune-modulating activities [67]. Oridonin also has a well-known potent anticancer activity against a wide range of cancer cell types, including prostate [34] and breast cancers [35] as well as acute leukemia [36]. Oridonin has been reported to exert antiproliferative effects on lung cancer cell lines in vitro and in vivo [37,68]. However, the effect of orodonine as a radiation adjuvant on lung cancer cells remains poorly understood. An ideal radiosensitizer increases the ability of radiation to kill cancer cells without changing the radiation response of healthy tissues [38]. In vitro models of cancer showed anti-tumor activity before and after irradiation by suppressing the proliferation of lung cancer cells in a time- and dose-dependent manner. Oridonin enhances the radiosensitization of lung cancer cells in all tested Gy doses and promotes apoptosis by Bax up-regulation and Bcl-2 down-regulation [39] (Figure 1). In the above study, oridonine used before irradiation significantly increased the Bax/Bcl-2 ratio, suggesting that apoptosis was promoted by the combination of oridonin and irradiation, compared to either single treatment [39]. Active Bax is recruited into the mitochondria and forms pores, leading to cytochrome c release [40]. Thus, it was hypothesized that the oridonine and irradiation combination activated the mitochondrial-mediated signaling pathway of apoptosis.

Additionally, oridonin facilitates radiation-induced ROS production, which causes DNA damage and induces apoptosis, when compared with ionizing radiation [41].

In a study by Park et al., oridonin has been shown to have a strong anticancer effect that inhibits the proliferation and clonogenic capacity of cancer cells. At the same time, researchers also identified cytotoxicity to non-cancerous cells that had not been considered in previous studies [41]. In particular, oridonin showed cytotoxicity to both normal lung epithelial cells and lung cancer cells. To reduce the toxic side effects of oridonine, a lower concentration of oridonine (5 µM) was used; a weaker anticancer effect was nevertheless observed compared to previous studies. The combination of 5 µM oridonin and radiation significantly increased ROS production, DNA damage, and apoptotic death in H460 cells. The obtained results suggest careful consideration of dose reduction to minimize side effects while promoting therapeutic efficacy [41]. Oridonin also suppresses epithelial-mesenchymal transition and migration by up-regulating E-kadherin and down-regulating some factors: vimentin, snail, and slug [42] (Figure 1). In an in vivo model, it was noticed that oridonin combined with radiotherapy intensified inhibition of tumor growth after 7 days of treatment as compared to using agents separately. It was observed that combination with oridonin induced up-regulation of caspase-3, γ-H2AX, markers of DNA damage, and apoptosis [41]. Oridonin is also able to induce the same adverse events. Some reports suggested hepatotoxicity. During the treatment, some serum parameters associated with liver toxicity increased, for example, alanine aminotransferase (ALP), alanine aminotransferase (ALT), and aspartate aminotransferase (AST). Although lactate dehydrogenase (LDH) decreased as compared to the control group [42], Furthermore, oridonin might affect the cytochrome p450 that interacts with some drugs. In vitro and in vivo expression of the CYP3a and CYP2c families significantly increased [69,70].

### 4.2. Celastrol

Tripterygium Wilfordii Hook F. has a long history of treating various disease conditions in traditional Chinese medicine. In the past, it was mainly used for the treatment of rheumatoid arthritis; however, nowadays, a wide range of its properties are still being explored. Many researchers have noticed anti-inflammatory, neuroprotective, anti-obsessive, and anti-artherosclerosic properties. One of the most active compounds isolated from Tripterygium is quinine methide triterpenoid-Celastrol. The anti-cancer properties of this compound in various in vitro and in vivo models are well-documented. Celastrol is able to regulate cancer’s functions including metastasis, proliferation, angiogenesis, and apoptosis. Moreover, celastrol directly affects both chemo- and radioresistant cancer cells, enhancing their sensitivity to chemo- or radiotherapy by various mechanisms [71]. Studies on radioresistant lung cancer cells in vitro showed that the use of celastrol combined with radiotherapy enhanced the radiosensitivity of cancer cells by suppressing proliferation and tumor growth. Respectively to these results, we examined potential factors inducing radiosensitivity such as EGFR, ErbB2, survivin, and Akt. Playing the role of an Hsp90 inhibitor, Celastrol remarkably decreased levels of EGFR, ErB2, and survivin, but Akt levels stayed constant. In addition, Celastrol induced apoptosis by up-regulating P53 [72], potentiating cytochrome c release from mitochondria, and cleaving caspases such as caspase-3, caspase-8, caspase-9, and PARP. During IR, Celastrol stimulated ROS and inhibited the activity of antioxidative thiol proteins such as glutathione and thioredoxin reductase. In vivo models did not notice any adverse events [43]. Despite most studies suggesting a narrow therapeutic window for celastrol, it also has adverse effects including weight loss, hepatotoxicity, cardiotoxicity, and infertility [71]. Newer reports suggest using celastrol with other agents to reduce its toxicity, for example, erastin [44].

### 4.3. Tashinone

Tanshinone is a phenanthrenequinone compound isolated from the Chinese medicinal herb Salvia miltiorrhiza, also called Danshen. What is more, tashinone has more than ten kinds of monomers; however, two of them, tashinone I (Tan I) and tashinone IIA (Tan IIA), are the most important in practice. Danshen has been used in the treatment of inflammatory diseases, including cerebrovascular and vascular diseases [73]. As a group of tanshinone compounds with a variety of pharmacological effects, they were considered anti-tumor agents. Studies showed promising effects on a wide variety of cancers, including lung cancer, prostate cancer, leukemia, liver cancer, and others [74]. Both compounds stimulate pro-apoptotic proteins such as p53 and Bax p-21 and induce apoptosis caspase, while simultaneously suppressing the production of anti-apoptotic proteins such as Bcl-2, c-Myc, and others.

Based on studies by Lee et al., it was found that in malignant pleural mesothelioma cells, Tan I induces autophagic cell death by activating c-Jun N-terminal kinase (JNK) signaling [75]. In subsequent studies, it was proven that Tan I exerts a cytotoxic effect by arresting the cell cycle and inducing apoptosis in the H1299 non-small cell lung cancer cell line [76]. However, until now, little was known about the effects and mechanisms of action of tanshinone on radiation resistance in lung cancer. In a study by Yan et al. on radioresistant lung cancer cells (H358-IR and H157-IR), the natural compound Tan I was shown to sensitize lung cancer cells to ionizing radiation. Tan I was found to significantly downregulate phosphoribosyl pyrophosphate aminotransferase (PPAT) expression by directly docking the protein’s PPAT active pocket, thus attenuating PPAT carcinogenic expression in radioresistant lung cancer cells [77].

### 4.4. Diosmetin

Diosmetin is a natural flavonoid compound found in legumes and olive leaves, showing a variety of pharmacological activities [78]. The pharmacological action of diosmetin has been found to have antimicrobial, antioxidant, anti-inflammatory, and anticancer effects. Diosmetin has recently been reported to exert its anticancer effects by up-regulating ROS levels and inhibiting Nrf2 [79,80]. In non-small cell lung cancer cells, diosmetin does not alter NRF2 mRNA transcription but disrupts Nrf2 stability through Keap1-mediated proteasomal degradation [80]. Based on their results, Androutsopoulos et al. found that diosmetin can induce antitumor activity in MCF7 breast cancer cells by bioactivating cytochrome P450 [45]. However, other researchers have suggested that diosmetin may induce G1/S phase arrest of the cell cycle and cell apoptosis in human A549 lung cancer cells [46].

However, one report has been published so far regarding diosmetin as a radiosensitizer to improve radiosensitivity. The mechanisms underlying diosmetin in combination with radiotherapy in the treatment of lung cancer cells remain to be fully elucidated. Research conducted by Xu et al. showed evidence that diosmetin can induce G1 phase arrest and thus increase the radiosensitivity of radioresistant A549/IR lung cancer cells [81]. In addition, diosmetin was also found to limit the repair of IR-induced DNA damage by inhibiting the activated Akt signaling pathway. Moreover, the combination of diosmetin and radiation has been shown to slightly enhance the killing effects of A549 cells, while diosmetin significantly promotes its sensitization to radiotherapy in A549/IR cell lines. This experiment confirms that diosmetin can effectively induce G1 phase arrest and ultimately sensitize A549/IR cells to radiation [81]. Taken together, these observations suggest that the natural compound diosmetin may act as a potential drug for the treatment of radioresistant lung cancer cells.

### 4.5. Schinifoline

Schinifoline is a 4-quinone alkaloid isolated from *Zanthoxylum schinifolium*, an aromatic shrub that commonly occurs in China, Japan, and Korea. Traditionally, *Zanthoxylum* has been used as a spice and medicinal herb with a variety of pharmacological effects. In East China, it has been used to treat symptoms of poisoning such as diarrhea, vomiting, and abdominal pain associated with various parasitic infections. In addition, external applications have been used to treat eczema [82]. Nowadays, the pharmacological properties of *Zanthoxylum* are still being explored. New research has provided more evidence on its anti-bacterial, anti-fungal, anti-insecticidal, anti-inflammatory, and analgesic properties [83,84]. Furthermore, recent studies have shown more evidence regarding the anticancer activity of *Zanthoxylum* compounds. Previous reports have suggested an impact on NF-κB inhibition, one of the most important immune factors [85]. Park et al. proved that the ethanol extract of *Zanthoxylum* leaves affects bladder cancer cell cultures through the induction of apoptosis. The presence of *Zanthoxylum* extract enhances the expression of pro-apoptotic factors and proteins, such as Fas and FasL ligands, and also activates pro-apoptotic caspases. Moreover, extracts increase the expression of cytochrome c by MMP suppression related to an increased Bax:Bcl-2 ratio. Finally, *Zanthoxylum* extract potentiates ROS production and inhibits the PI3K/Akt signaling pathway [47] (Figure 1). However, less is known about schinifoline’s pharmacological activities separately. Wang et al. investigated whether schinifoline enhanced ionizing radiation effects on human non-lung small cancer cells. In that study, schinifoline indicated pro-cytotoxicity properties in A549 cells by suppressing cell proliferation in monotherapy or in combination with IR. A radiosensitizing effect on cancer cells was observed additionally on clonogenic assay exposure at different doses of the IR fraction. Schinifoline in low concentrations affects IR used in various doses, causing the radiosensitization of cancer cells. The compound, in combination with an increased dose of IR, decreased the percentage of survival fraction. Moreover, flow cytometry showed that A549 cells treated with schinifoline before irradiation potentiate the G2/M phase and cause apoptosis [48]. This study was the only one focused on schinifoline’s anticancer properties as a radiosensitizer.

### 4.6. Huachansu

Another interesting traditional Asian medicament is cinobufacini, also called chansu in Chinese and senso in Japanese. The main source of this component is dry toad skin from bufo bufo gargarizans Cantor. The injectable water-soluble extract preparation used in treatment is called Huachansu. For thousands of years, chansu has been used for the treatment of various diseases such as a lenitive, cardiotonic, antimicrobial, or anesthetic agent [49]. Moreover, it has been used for the treatment of heat and toxins syndrome including severe symptoms of gastroentitis, vomiting, diarrhea, and fever [50]. In Chinese, Huachansu has a special oncology space. Huachansu is approved by the Chinese State Food and Drug Administration to use in the treatment of a wide spectrum of cancers, especially pancreatic, lung, and colorectal cancer [50]. The major group of compounds found in Huachansu are C-24 steroids called bufadienolids (including bufalin, cinobufagin, resibufogenin, telcinobufangin), which possess cardiotonic properties and has the same mechanism of action (Na^+^, K^+^-ATPase inhibitors) as digoxin [49]. In recent years more evidence has been found regarding the anti-cancer mechanism of Huachansu components in action. A great number of studies have provided information relating to how the active compounds of Huachansu potentiate anti-tumor effects through a decrease in proliferation, cell-cycle suppression, inhibition of angiogenesis, and differentiation and induction of apoptosis [50]. At the molecular level, it is known that compounds of Huachansu act selectively in cancer cells to induce cytotoxic effects by enhancing ROS level, which leads to DNA damage response, and results in the activation of the G_2_/M checkpoint due to apoptosis [51]. Furthermore, data suggest that Huachansu in the complex with chemotherapy intensifies cytotoxic effects and improves patients’ conditions [52,53,86]. A meta-analysis in clinical trials showed the potential in strengthening chemotherapy through a statistically significant tumor response, one-year survival, Karnofsky performance status, pain alleviation, and reduction of adverse effects [87]. However, Huachansu’s impact on radiotherapy effectiveness is not well-known. Some data indicate that compounds of Huachansu potentiate ionizing radiation effects, for instance, in esophageal or lung cancer cells, by changing the cell cycle schedule, inducing DNA double-strand breaks through time suppression DNA repair mechanisms, and sensitizing cancer cells to radiation-induced apoptosis [54,88]. In lung cancer, Huachansu extends the attendance of γH2AX foci associated with DNA impairment and changes the proportion of pro- and anti-apoptosis factors (Figure 1). The synergism between radiation and Huachansu forces the up-regulation of cleaved caspase-3 and cleaved poly-(ADP-ribose) polymerase (PARP) that simultaneously inhibits the activity of Bcl-2 and P53 proteins [88]. Moreover, bufalin suppresses the activity of TopoII, which causes delayed DNA repair and leads to the radiosensitivity of cancer cells [89].

## 5. Conventional Medicaments as Radiosensitizers in Off-Label Use

More often, drug discovery and development focus on the utilization of well-known drugs for other conditions. Off-label use is beneficial for many reasons—therapeutic doses and adverse effects are well-documented. What is more, testing drugs already used in medicine significantly reduces the costs associated with drug development departments. Many of them are low-priced and have a long history of use in chronic diseases.

### 5.1. Antidiabetic Drugs

#### 5.1.1. Metformin

Metformin is a dimethyl biguanide derivative usually used as a first-line treatment in diabetes mellitus type II. It is related to the dose-dependent ability for the regulation of glucose metabolism. Metformin inhibits gluconeogenesis and glycogenolysis processes in hepatic cells, reduces glucose absorption in bowels, and enhances glucose uptake to tissues [90]. At the molecular level, metformin acts by inhibiting the mitochondrial electron transport chain at Complex I (NADH: ubiquinone oxidoreductase), which leads to the reduction of intracellular ATP and enhancement of AMP concentration, which activates AMPK. The upregulation of AMPK inhibits enzymes that take part in fatty-acids biosynthesis, as well as gluconeogenesis. Furthermore, metformin stimulates fatty acid oxidation and glucose transport proteins [90,91].

The pharmacological effect of metformin is not limited to its hypoglycemic properties. The anti-proliferative effect of this compound is well-known, especially since diabetes mellitus type II is characterized by an increased risk of cancer, especially colorectal, endometrial, and postmenopausal breast cancer [55,92]. Zarrouk et al. indicated that metformin inhibits mTOR (the mammalian target of rapamycin complex I) and also suppresses expression of the transcription factors c-Myc and Hif-1α (hypoxia-inducible factor 1 alpha) (Figure 1). Generally, metformin regulates metabolic reprogramming, leading to the control of proliferation and antigen-activated T cell growth [93]. The anti-tumor activity of metformin is also well-documented in various clinical trials [94]. Especially as an adjuvant, it has a special place in colorectal, prostate, pancreatic, lung, or breast cancer therapy [95,96]. The anti-cancer activity of metformin is related to its basic function as a hypoglycemic agent. In cancer cells, metformin acts through two different mechanisms: directly by AMPK activation and the mTOR pathway, or indirectly by regulation of angiogenesis, and in fibroblasts through regulation of immunosuppression, changing the tumor environment. What is more, the literature gives evidence that metformin potentiates cytotoxic effects in combination with chemotherapeutics and radiotherapy. Focusing on the role of metformin in radiation therapy, many clinical trials suggest an influence on tumor response and an increase in survival rate [97]. In many cancers, metformin preferentially interacts with cancer stem cells by potentiating the genotoxic effect of radiation [56,98].

Anyhow, knowledge about metformin as radiosensitizers is still growing. Generally, in various cancers, metformin reprograms cell metabolism, causing tumor oxygenation thus improving radiotherapy efficacy [99]. In lung cancer, Storozhuk et al. noticed that metformin combined with IR maintains the activation of ATM–AMPK–p53/p21^cip1^ and inhibition of Akt–mTOR–4EBP1 pathways, and causes the inhibition of angiogenesis and induction of apoptosis [57]. Subsequent research by Riaz et al. with cisplatin confirmed these results [58]. Another variant proven by Sun et al. includes the role of metformin as a radiosensitizer through a KEAP1-independent mechanism which causes a decrease of NRF2 due to ubiquitination and proteasomal degradation [100].

What is important, metformin also attenuates pulmonary fibrosis and pneumonitis, which are side effects of radiation therapy [59,101].

Besides metformin, other biguanide phenformins possess similar radiosensitive effects on cancer cells, as tested in a lung cancer model [102].

#### 5.1.2. Rosiglitazone

Rosglitazone represents thiazolidinediones (TZDs), agonists of the nuclear receptor peroxisome-proliferator-activated receptor (PPAR), that are used in clinical practice to improve glucose tolerance and insulin sensitivity in patients with obesity, hypertension, etc. [103]. However, rosiglitazone’s pharmacological effects may be beneficial in different areas. PPARγ overexpression occurs in various types of cancer, so rosiglitazone is one of the potential anti-cancer candidates [60]. Rosiglitazone is well-studied in various cancer models such as lung cancer, pancreatic cancer, colon cancer, renal cancer, glioma, leukemia, and others [61,62,65,104,105,106,107,108]. In a murine model, rosiglitazone prevents the progression of preinvasive lung cancer [66]. Furthermore, anti-cancer activity includes induction of pro-apoptotic factors such as TNF-α, suppression of tumor growth by blocking TGF-β, invasion, metastasis, cell cycle arrest, modulation of gene activity, ROS production, and reduction of cancer stemness in vitro and in vivo [61,62,65,104,105,106,107,108]. Moreover, rosiglitazone stimulates anti-cancer properties of various cytostatics, for instance, doxorubicin, cisplatin, and carboplatin, as well as adverse effects reduction such as cardiotoxicity, nephrotoxicity, and immunosuppression [109,110,111,112]. Several studies indicate that rosiglitazone regulates ABC transporters that are responsible for multidrug resistance [60,61,62,65,66,104,105,106,107,108,109,110,111,112,113,114]. In the lung cancer model, rosiglitazone sensitizes lung cancer cells to carboplatin [115]. Generally, most of the research focused on rosiglitazone and radiation therapy is involved with radioprotective properties in healthy tissues that exert inhibition of inflammation and toxicity through regulation of TGF-β, the NF-κB p65 subunit, and genes associated with TNF-α, Il-6, and Il-1β [116,117,118,119]. Nevertheless, radiosensitizing effects were described in a few cancer models, for instance, colorectal cancer, cervical cancer, pancreatic cancer, and lung cancer [120,121,122,123]. There are only two studies about PPARγ radiosensitizing properties in lung cancer, and one of them was rosiglitazone. Kaur et al. observed that rosiglitazone combined with IR was able to stimulate radiation-induced bax-mediated cell death in the radioresistant NSCLC A549 cell line. Moreover, the treatment also inhibited radiation-induced AKT (Protein Kinase B) phosphorylation, and radiation-induced apoptosis was inversely related to TP53 levels [124].

With other PPARγ agonists, rosglitazone taken together with IR synergistically induced DNA damage and apoptosis, which were regulated by ROS in vitro [125]. More precisely, Han et al. observed caspase-3-mediated poly (ADP-ribose) polymerase (PARP) cleavage, caspase-8 activation, and also γ-H2AX expression, all associated with DNA damage [126].

### 5.2. Antiartheroslerotic Drugs

#### 5.2.1. Lovastatin

Lovastatin is one of the HMG-CoA reductase inhibitors and one of the most important drugs used in hypercholesterolemia. Taking into consideration their common use in clinical practice (every fourth person above 40 takes statins to improve lipid levels) and low price, lovastatin, and other statins constitute objects of research in other diseases [124]. A lot of studies indicate that statins may have potential as anti-cancer agents in two ways: as drugs for prevention and treatment. In a population-based cohort study, Cardwell et al. suggested that cancer patients who took statins before diagnosis had a reduced cancer mortality rate [60]. The anti-cancer activities of lovastatin were documented in breast, gastric, liver, colon, thyroid, ovarian, lung, and other cancers [63,64,127,128,129,130,131]. The mechanism of the anti-cancer activity is not fully understood. Generally, blocking the HMG-CoA pathway enhances L-mevalonate levels, which are precursors of isoprenoids, which play a role in apoptosis. In lung cancer models treated with lovastatin, a lower level of glutathione and an increased level of cytochrome c, caspase-3, p53, and Bax protein levels were observed [126]. On the other hand, Walther et al.’s study indicated other mechanisms of cancer cell death via COX-2 activation and PPAR [132]. Moreover, lovastatin combined with other anticancer agents such as doxorubicin and paclitaxel improves their properties [129,130,131,132,133]. However, lovastatatin’s influence on breaking radioresistance is not fully known.

Lovastatin also influences radiotherapy effectiveness. Sanli et al. proved that the combination of radiotherapy with lovastatin enhances lung cancer cell death in a dose-dependent manner, compared to radiotherapy alone. Furthermore, immunoblotting shows that IR with lovastatin inhibits Akt but stimulates Erk phosphorylation. Additionally, statin enhances basal and IR-induced AMPK phosphorylation. Lovastatin blocks the expression of p53 protein, kinase inhibitors (p21^cip1^ and p27^kip1^) induced by IR and predominance cell phase from G1-S phase or G2-M phase into apoptosis [134] (Figure 1).

What is important is that lovastatin does not induce DNA double-strand breaks in normal epithelial cells but acts radioprotectively [135]. It also restrains pro-inflammatory reactions and fibrosis in normal tissue [136]. A recent study in a lung cancer model showed that lovastatin interacts with Rho small GTPases to protect lung epithelial, fibroblast, and microvascular epithelial cells against IR-induced toxicity through improved repair DNA mechanisms and caspase-dependent apoptosis [137].

#### 5.2.2. Simvastatin

Similar to Lovastatin, other compounds belonging to the HMG-CoA reductase inhibitor class show the potential to improve radiation therapy results. A lot of research has indicated that simvastatin acts radiosensitively in various cancer models such as head and neck cancer, esophageal cancer, myeloma, colorectal cancer, lung cancer, myeloma, breast cancer, and prostate cancer [138,139,140,141,142,143,144]. In colorectal and prostate cancers, it was observed that simvastatine acts antiproliferative in a dose-dependent manner, increases pro-apoptotic factors such as cleaved caspase 3, and inhibits phosphorylation of ERK1/2 [143,144]. Additionally, flow cytometry analysis of myeloma cancer cells after irradiation indicated increased ROS production, arrest of cells in G0/G1 and G2/M phases, decreased antioxidant enzymes, decreased intensification of P53, Bax, and Caspase 3 gene expression, and a decrease in Bcl2 [139]. A few studies also suggest a simvastatin combination with IR for cancer stemness [140,141]. Yu et al. checked simvastatin efficacy with IR in vitro and alone using an MTT assay and western blot analysis technique. They noticed that simvastatin inhibits the proliferation of the Lewis cell line by inhibiting the activity of p38 and the expression of p-p38. What is important is that simvastatin reduced the invasion and motility of the Lewis cell line by down-regulating the expression of RhoA and MMP-2. In combination with radiotherapy, simvastatin can inhibit tumor growth and metastasis and synergistically improve the treatment efficacy of radiotherapy [145]. In vivo, simvastatin in combination with IR acts radioprotectively by reducing multiple RILI (radiation-induced lung injury) indicators, including vascular leak, leukocyte infiltration, and histological signs of oxidative stress. Simultaneously, it participates in reversing RILI-associated dysregulated gene expression, including p53, nuclear factor-erythroid-2-related factor, and sphingolipid metabolic pathway genes [142]. Simvastatin alone may be beneficial in lung cancer through the regulation of genes responsible for cancer progression, such as down-regulation of METTL3, in regulating relative levels of EZH2 and EMT-associated genes, as well as in metastatic abilities [146].

### 5.3. Anti-COX Drugs

#### Nimesulide

Nimesulide is a medicine belonging to the non-steroidal anti-inflammatory drug (NSAIDs) group. Similar to others, nimesulide is mainly used to relieve pain, reduce inflammation, and bring down a fever. Nimesulide inhibits the production of prostaglandins due to their selective inhibition of cyclooxygenase COX-2 [89]. Importantly, several studies observed high levels of COX-2 in tumors that contribute to invasion and ultimately, poor survival [147,148,149,150]. Prostaglandins are essential for the tumor microenvironment where they stimulate various pathways associated with endothelial cell migration, angiogenesis, and immunosuppression [151]. Therefore, NSAIDs constitute one of the potential candidates in the treatment of various cancers.

Anti-proliferative properties were examined in vitro in various models of cancers including pancreatic cancer, breast cancer, colorectal cancer, and lung cancer [152,153,154,155]. Hida et al. demonstrated anti-proliferative properties in a dose-dependent manner and induction of apoptosis. Nimesulid’s responsiveness was dependent on COX-2 level [156]. Afterwards, Han et al. suggested that nimesulide induces apoptosis and inhibits the proliferation of non-small cell lung cancer by inducing the expression of the p21 gene through MEK-1/ERK signaling and through DNA—protein interactions [155].

Otherwise, NSAIDs fulfill a role as a preventive anticancer agent. Olsen et al., in their comparative study, suggested the preventive role of NASIDs for smokers to reduce lung cancer risk [157].

However, a few research works mentioned the radiosensitive properties of nimesulide and its analogue. Hong et al. proved that nimesulide enhances the radiotherapy effect by decreasing the survival fraction in clonogenic survival assay. When compared to celecoxib, nimesulide has higher radiosensitive efficacy. Apoptosis significantly increases in samples exposed to nimesulid in comparison to samples only exposed to radiation or nimesulide alone. What is more, higher levels of caspase-8 and caspase-3 were observed [158]. Another study showed that the nimesulide analog, NS-398, possesses similar properties to nimesulide. NS-398 decreased the survival fraction of lung cancer cells in a concentration-dependent manner, thereby enhancing the pro-apoptotic properties of radiation therapy. Additionally, the combination of analog with IR delayed tumor growth [159].

## 6. Conclusions

Lung cancer is a serious epidemiological challenge. An increasing rate of this type of cancer and mortality necessitates the search for new therapies to improve the effectiveness of oncological treatment. Taking into consideration various factors of carcinogenesis, oncologists need to use target therapy, which is tailored to the patient. Radiotherapy is used in conventional therapy to treat various types of cancers in different locations, but similar to other therapies, have some limitations.

Increasing radioresistance forces scientists to examine new methods of cancer treatments and improvements to the response to radiotherapy. The wealth of Chinese medicine is a basis for further perspectives on drug development. On the other hand, well-known drugs are also verified in other indications. The advantages such as low costs, well-known adverse events, and better-known pharmacokinetics and pharmacodynamics encourage further research in various clinical trials.

This review shows potential candidates how to improve radiotherapy effectiveness in a lung cancer model. The compounds of natural origin used in traditional Chinese medicine, such as Oridonin, Celastrol, Schinifoline, Diosmetine, and Huachansu, showed pro-ionizing radiation effects. Albeit, each of these was examined in a small number of research works. For a more comprehensive study, a wider scope of research, as well as clinical trials, may be considered in the future. The off-label use of Metformin, Lovastatin, and Nimesulide indicates their pro-radiosensitizing effects on tumor cells. However, as with every drug, they may cause adverse effects. Thus, the off-label use of Metformin, Lovastatin, and Nimesulide combined with radiotherapy may be considered, taking into account the balance of benefits and losses.

## Figures and Tables

**Figure 1 ijms-24-11206-f001:**
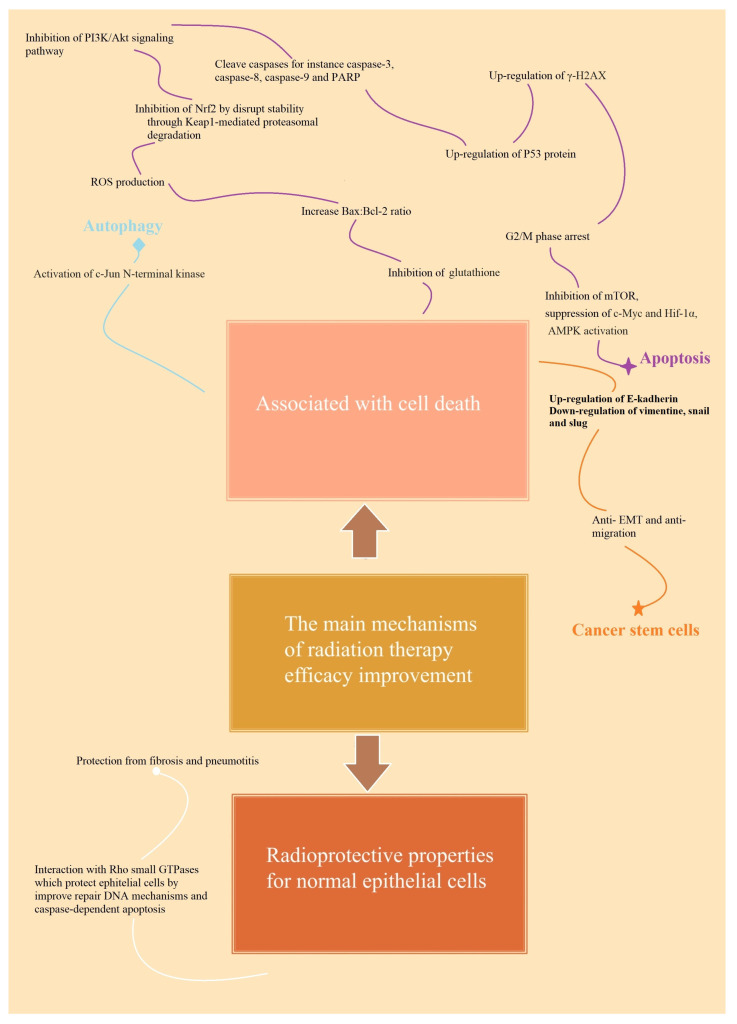
The main mechanisms of radiotherapy response improvement due to active compounds. The figure explains the main mechanisms of TCM and conventional compounds, divided into two sections: cell death and protective properties. All substances reviewed possess pro-apoptotic functions. Metformin, rosiglitazone, lovastatin, and simvastatin act radioprotectively. Tashinone may induce autophagy, whereas oridonine suppresses cancer stemness.

**Table 1 ijms-24-11206-t001:** Compound’s mechanism of action alone and in combination with radiation therapy.

Substance	Mechanism of Action	References
Oridonin	Apoptosis induction: Up-regulation bax, down-regulation of bcl-2, potentiate radiation-induced ROS production, up-regulation of caspase-3, γ-H2AX Anti-EMT and migration: up-regulation of E-kadherin, down-regulation of vimentin, snail, slug	[24,25,26,27]
Celastrol	As Hsp90 inhibitor: decreases levels of EGFR, ErB2 and surviving Apoptosis induction: up-regulation of P53 protein, potentiated cytochrome c release from mitochondria, cleaved caspases for instance caspase-3, caspase-8, caspase-9 and PARP	[31,32]
Tashinone	Apoptosis induction: p53, Bax p-21, suppressed production of anti-apoptotic proteins such as Bcl-2, c-Myc Autophagy induction: c-Jun N-terminal kinase activation With IR: PPAT downregulation	[34,35,36,37]
Diosmetin	Apoptosis induction: increasing of ROS level, inhibition of Nrf2 by disruption stability through Keap1-mediated proteasomal degradation, G1/S phase arrest of the cycle With IR: IR-induced DNA damage repair inhibition, inhibition of Akt signaling pathway, induction of G1 phase arrest	[38,39,40,41,42]
Schinifoline	Apoptosis induction: Production of Fas and FasL, pro-apoptotic caspases, cytochrome c, increased bax:Bcl-2 ratio, stimuli ROS production, inhibition of PI3K/Akt signaling pathway, G2/M arrest	[43,44]
Huachansu	Apoptosis induction: enhance of ROS level, activation of the G_2_/M checkpoint, extending attendance of γH2AX foci, up-regulation cleaved of caspase-3 and cleaved poly-(ADP-ribose) polymerase (PARP), inhibition of Bcl-2 and P53 protein, suppression of TopoII	[45,46]
Metformin	Apoptosis induction: inhibition of mTOR, suppression of c-Myc and Hif-1α, AMPK activation with IR: maintain activation of ATM–AMPK–p53/p21^cip1^ and inhibition of Akt–mTOR–4EBP1, through KEAP1-independent mechanism, causes decrease of NRF2 due to ubiquitination and proteasomal degradation, preferentially acts with CSC Other with IR: protect from fibrosis and pneumonitis	[47,48,49,50,51,52,53,54]
Lovastatin	Apoptosis induction: lower-level of glutathione, increase of cytochrome c, caspase-3, p53, bax, COX-2 activation and PPARγ with IR: Akt inhibition, stimulation of Erk phosphorylation, block expression p53 protein and kinase inhibitors p21^cip1^ and p27^kip^ Other with IR: in normal epithelial tissue acts radioprotective, interact with Rho small GTPases protect lung epithelial, fibroblasts, microvascular epithelial cells against IR-induce toxicity by improved repair DNA mechanisms and caspase-dependent apoptosis	[55,56,57,58,59]
Rosiglitazone	Apoptosis induction: stimulation of bax-mediated cell death, inhibited radiation-induced AKT (Protein Kinase B) phosphorylation, radiation-induced apoptosis was inverse-related with TP53 levels Radioprotective function: regulation of TGF-β, NF-κB p65 subunit, and regulation of genes associated with TNF-α, Il-6 and Il-1β	[60,61,62]
Simvastatin	Inhibition of tumor growth and metastasis, reduction of invasion and motility of Lewis cell line through down-regulating the expression of RhoA and MMP-2. Other with IR: improvement of radiation-induced lung injury through genes regulation (p53, nuclear factor-erythroid-2-related factor, and sphingolipid metabolic pathway)	[63,64]
Nimesulide	Apoptosis induction: induction of the expression of the p21 gene through MEK-1/ERK signaling and also DNA-protein, associated with COX2 level With IR: increase of caspase-8 and caspase-3 level	[65,66]

## Data Availability

The data presented in this study are available upon request from the corresponding author.

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
