# Peer review of "A Cross-Talk about Radioresistance in Lung Cancer—How to Improve Radiosensitivity According to Chinese Medicine and Medicaments That Commonly Occur in Pharmacies"

_ijms, 2023, doi:10.3390/ijms241311206_

Round 1

Reviewer 1 Report

The Review entitled "A cross talk about radioresistance in lung cancer. How improve radiosensitivity according to Chinese Medicine and medicaments commonly occur at Pharmacy" by Paulina Nowak, Iwona Bil-Lula, Mariola Śliwińska-Mossoń essentially clarify two critical points: the radiosensitizers with natural origin used in traditional Chinese medicine and second well- known drugs used in conventional medicine. The Review is well written and very interesting, however I think it could be improved.

1. Authors should describe in more detail the molecular and cellular characteristics of SCLC and NSCLC, with particular regard to the main mutated genes. Authors can draw up a separate paragraph.

2. Do Oridonin, Celastrol, Tashinone, Diosmetin, Schinifoline, Huachansu  act at the level of a precise molecular pathway? It is requested to describe it more analytically, especially the target genes. Eventually, the authors could synthesize the molecular targets in a table.

What do the authors think is the best compound? And why?

3. Regarding the drugs used in metabolic syndrome, have only the radioxytivizing effects of metformin and lovastatin been described? What is known about rosiglitazone, sufanureas and for example simvastatin which are widely used. Please read more about this paragraph.

4. Could the effect of conventional drugs as radiosensitizers in off-label use be enhanced by the main molecules used in Chinese medicine?

5. The authors could elaborate a graphical abstract in which they summarize the discussion points of the review by bringing out the conclusions with immediate visual impact.

Minor editing of English language required

Reviewer 2 Report

The review wrote by Nowak et at. is valid: radiotherapy treatment is in extreme need of alternative radiosensitizers against cell resistance.

My comments are as follows:

- Please mention in the introduction the common radiosensitizers currently used in combination with radiotherapy

- Format of Figure 1 has to be extensively reviewed (with connection to the reported drugs)

- The mechanism of action of most of the drugs is nicely described in both table 1 as well as in the text. However, please explain the current preclinical/clinical phase of the reported drugs, including advantages and limitations. This will help to correctly classify each of them and understand their potential.

Please, correct especially the passive forms of the verbs and the use of articles. 

Round 2

Reviewer 1 Report

The Review was significantly improved by the authors, so it is now publishable on ijms.

Minor editing of English language required

Reviewer 2 Report

I thank the authors for the revision and the rebuttal letter. Figure 1 is now clearer and the rest of the manuscript has highly improved. It is now a comprehensive review and ready to be published.

I have only one last and minor comment:

As also suggested by the 2nd reviewer, I would add in the manuscript (either at the end of paragraph 4 or 5, or in the conclusions) your comment on what you think is the best/preferred radiosensitizers. Please do that for both, the Chinese derived- compounds and the off-labels medicaments. 

I believe this is highly important for the community working on similar compounds or for future preclinical/clinical translation. This will also answer one of my previous points in the first round of the review.

Please, check-out for remaining minor English mistakes that are still present in the manuscript